

# Anoxic conditions maintained high phosphorus sorption in humid tropical forest soils

Yang Lin, Avner Gross[‡], Christine S. O'Connell, Whendee L. Silver

Department of Environmental Science, Policy, and Management, University of California, Berkeley, CA 94720, USA
5 [‡]Current address: Department for Geography and Environmental Development, Ben Gurion University of the Negev, Israel.

*Correspondence to*: Yang Lin (yanglin@berkeley.edu)

**Abstract.** The strong phosphorus (P) sorption capacity of iron (Fe) and aluminum (Al) minerals in highly weathered, acidic soils of humid tropical forests is generally assumed to be an important driver of P limitation to plants and microbial activity in these ecosystems. Humid tropical forest soils often experience fluctuating redox conditions that reduce Fe and raise pH. It is 10 commonly thought that Fe reduction generally decreases the capacity and strength of P sorption. Here we examined the effects of 14-day oxic and anoxic incubations on soil P sorption dynamics in humid tropical forest soils from Puerto Rico. Contrary to the conventional belief, soil P sorption capacity did not decrease under anoxic conditions, suggesting that soil minerals remain strong P sinks even under reducing conditions. Sorption of P occurred very rapidly in these soils, with at least 60% of the added P disappearing from the solution within six hours. Estimated P sorption capacities were one order of magnitude 15 higher than the soil total P contents. However, the strength of P sorption under reducing conditions was weaker, as indicated by the increased solubility of sorbed P in $NaHCO_3$ solution. Our results show that highly weathered soil minerals can retain P even under anoxic conditions, where it might otherwise be susceptible to leaching. Anoxic events can also potentially increase P bioavailability by decreasing the strength, rather than the capacity, of P sorption. These results improve our understanding of the redox effects on biogeochemical cycling in tropical forests.

**Keywords**: Luquillo CZO and LTER, phosphorus availability, nutrient limitation, sorption kinetics, sorption isotherms, sorption rate, anaerobic, aerobic, redox oscillation

## 1 Introduction

Phosphorus (P) is often thought to limit net primary productivity and organic matter decomposition in humid tropical forests 25 that grow on strongly weathered soils (Vitousek and Sanford Jr., 1986; Cleveland et al., 2011; Camenzind et al., 2017). In these soils, geochemical reactions of adsorption and precipitation, also known as sorption, directly compete with plant roots and microorganisms for phosphate (Thompson and Goyne, 2012). Sorption reactions can immobilize a large of amount of P that can exceed the actual size of the labile soil P pools (de Campos et al., 2016; Roy et al., 2017; Gross et al., 2018) at the scales of seconds to hours (Olander and Vitousek, 2004; McGechan and Lewis, 2002). The sorbed P is generally not readily





available for plant and microbial uptake (Tiessen and Moir, 1993). Thus, mineral sorption plays a key role in constraining the biological availability of P in these ecosystems (Johnson et al., 2003; Reed and Wood, 2016).

How do plants and microbes acquire P in tropical forest soils with high P sorption potential? One of the commonly
hypothesized, yet rarely tested, mechanisms is that reducing events can decrease the effectiveness of soil minerals in sorbing P (Chacón et al., 2006; Lin et al., 2018). These soils often contain high concentrations of redox-sensitive poorly crystalline or amorphous Fe minerals (Hall and Silver, 2015; Wilmoth et al., 2018; de Campos et al., 2016). Under anoxic conditions, Fe(III) minerals undergo reductive dissolution and are transformed into Fe(II) phases that are assumed to be less effective at binding P (Quintero et al., 2007; Rakotoson et al., 2014; Shenker et al., 2005). Reducing conditions increase pH and decrease surface
charges that are thought to be central to P sorption (Oh et al., 1999; Willett, 1989). Although Al minerals, which are often enriched in these soils, are not redox-active, redox-induced pH changes can control the speciation of Al minerals and consequently affect P sorption (Haynes, 1982; Gustafsson et al., 2012).

Redox effects on soil P sorption processes have not been well studied in tropical forest soils, even though periodic redox
oscillations are well documented in these environments (Barcellos et al., 2018; Schuur et al., 2001; Chen et al., 2018; Keiluweit et al., 2016; Wieder et al., 2011; O'Connell et al., 2018; Silver et al., 1999). Some past studies from tropical ecosystems have reported increases in extractable soil P during anoxic events (Chacón et al., 2006; Peretyazhko and Sposito, 2005; Maranguit et al., 2017), which is consistent with the hypothesis that anoxic conditions weaken P sorption, although the mechanisms remain unclear. To better understand the importance of redox events in regulating P bioavailability in tropical forest soils, it is
necessary to examine both the capacity and strength of P sorption. Sorption capacity is usually characterized by the maximum amount of P that soil minerals can sorb and can be measured using the sorption isotherm method (McGechan and Lewis, 2002). The strength of P sorption is related to the potential bioavailability of the sorbed P, as strongly sorbed P is considered to be largely unavailable plant roots and microbes. It can be evaluated by measuring the solubility of sorbed P in various solvents such as $NaHCO_3$ and NaOH solution (Ryden and Syers, 1977).


We determined the effects of redox conditions (oxic vs. anoxic incubations) on P sorption characteristics in soils from humid tropical forests in Puerto Rico. We conducted two types of P sorption experiments: 1) the standard sorption isotherm to evaluate the capacity of P sorption; and 2) the P sorption time curve to evaluate the rate of P sorption over time. These P sorption experiments were conducted using soils from two distinct parent materials with different Fe, Al, and P concentrations and two
topographic positions (frequently reduced valleys and more aerated slopes), allowing us to evaluate the effects of soil redox history on P sorption processes. We hypothesized that anoxic conditions would decrease soil P sorption capacity due to a decrease in Fe(III) concentrations and associated weakening of the Fe-P bond. We also hypothesized that redox-induced




decreases in P sorption capacity would be accompanied by increases in the solubility of sorbed P, indicative of weaker P sorption strength.

## 2 Material and Methods

### 2.1 Study sites and soil sampling

Soils were collected from humid tropical forests in the Luquillo Experimental Forest (LEF) in Puerto Rico, part of the NSF sponsored Long Term Ecological Research (LTER) and Critical Zone Observatory (CZO) networks, and the DOE sponsored NGEE Tropics program. The mean annual temperature decreased from about 23 °C at 350 m to about 19 °C at 930 m elevations above sea level (Weaver and Murphy, 1990; Brown et al., 1983). Mean annual rainfall increased from about 3,200 mm at 300 m to about 4,800 mm at 1000 m with no clear seasonal pattern (Murphy et al., 2017). Mineral soils (0-15 cm depth) were

collected from two sites, El Verde and Rio Icacos, featuring different parent materials and soil characteristics (Table 1). The El Verde site is located at approximately 380 m elevation within the Tabonuco forest zone. The Rio Icacos site is located at approximately 630 m within the Colorado forest zone, and characterized by more frequent cloud condensation and abundant epiphytes. Soils from El Verde were Hapludoxes and derived from volcaniclastic material (Scatena, 1989). Soils from Rio Icacos were Dystrudepts and developed from quartz diorite material, which contained approximately half as much total P

compared to volcaniclastic material and lower concentrations of Fe and Al oxides (Mage and Porder, 2012). As a result, soils from El Verde generally have higher total concentrations of P, Fe, and Al minerals (Mage and Porder, 2012; Coward et al., 2017). Topographic locations (ridge, slope, and valley) play a key role in determining soil redox conditions and biogeochemical processes (Scatena, 1989; Silver et al., 1994; O'Connell et al., 2018; Hall and Silver, 2015). In these highly dissected landscapes, catenas typically shift from well-aerated ridges to valley bottoms that experience frequent reducing events. Slopes

cover approximately 65% of the landscape, while ridges and valleys make up the rest in approximately equal proportions (Scatena and Lugo, 1995). For the sorption experiments, soils from both sites were collected from valley and slope positions after removing the surface litter layer. For the solubility experiment, we only analyzed the slope and valley soils from the El Verde site due to logistical limitations in accessing the Rio Icacos site after Hurricane Maria. The valley soil sampled for the solubility experiment was significantly drier than that sampled previously (Table 1). Soils were shipped overnight to the

University of California, Berkeley, at an ambient temperature and were immediately gently homogenized by hands upon arrival and visible plant debris, rocks, and macro-fauna removed.

### 2.2 Redox treatments

To examine the effects of low redox conditions on soil P sorption processes, soil samples were pre-incubated under anoxic or

oxic conditions for 14 days. Soils were subsampled in quart-size, glass jars with gas-tight lids (~ 100 g each, oven-dry



equivalent (ODE)). For those in the anoxic treatment, the jar headspace was evacuated and flushed with $N_2$ three times before being transferred to an anaerobic glove box (90% $N_2$, 8% $CO_2$, and 2% $H_2$; Coy Laboratory Products, Grass Lake, MI). Jars were sealed inside the glove box and vented every two or three days. Jars in the oxic treatment were sealed under an ambient atmosphere and vented following the same schedule as those from the anoxic treatment. Jars were stored in cardboard boxes

in the dark during the incubation. Soils from El Verde and Rio Icacos were analyzed in two separate campaigns following the same experimental approach due to space limitations in the glovebox.

## 2.3 Phosphorus sorption experiments

After pre-incubation, soil P sorption was evaluated in two ways. We used P sorption isotherm experiments to determine

sorption across a range of standard P loadings over 24 hours. We also conducted P sorption time curves that characterize the disappearance of solution P with one level P of addition at multiple time points over 48 hours (Henry and Smith, 2006). For P sorption isotherms, aliquots of 15 g ODE soil were subsampled into pint-size glass jars containing 150 ml of 0.01 M $CaCl_2$ to reach a soil to solution ratio of 1:10. The $CaCl_2$ solution had been previously spiked with 100 mg P $L^{-1}$ $KH_2PO_4$ stock solution to reach four levels of P concentrations (500, 1000, 5000, and 10,000 mg P $kg^{-1}$ soil). Preliminary trials showed that P addition

at lower concentrations (i.e., 10 and 100 mg P $kg^{-1}$ soil) resulted in near complete sorption within 24 hours. There was a total of 128 samples (4 replicates × 4 levels of P addition × 2 redox treatments × 2 topographic locations × 2 sample sites). Soil slurry was then amended with 1 mL of toluene to inhibit microbial activity before being manually shaken for one minute. Samples from the anoxic treatment were prepared in the anoxic glove box with degassed solutions. The slurry was manually shaken periodically to mix soils with solution. After 24 hours, 5 mL subsamples of soil slurry were extracted from each jar

and filtered through 0.45 μm syringe filters into test tubes and then acidified with HCl to a final $H^+$ concentration of 0.1 N to prevent the oxidation and precipitation of dissolved Fe(II).

For P sorption time curves, batch experiments were started by subsampling aliquots of 30 g soil (ODE) into quart-size glass jars with 300 ml of 0.01 M $CaCl_2$ solution with a single pulse of 1000 mg P $kg^{-1}$ soil and 2 mL of toluene. A total of 32 samples

(4 replicates × 2 redox treatments × 2 topographic locations × 2 sample sites) were prepared. Subsamples of suspended soil were taken after 5 minutes, 40 minutes, 2 hours, 6 hours, 12 hours, 24 hours and 48 hours following the previously described method. Solution P concentrations from the sorption experiments were determined colorimetrically following Murphy and Riley (1962).



## 2.4 Soil Fe and Al analyses and P fractions

Redox-active Fe pools were measured after the pre-incubation. Acid-soluble Fe was extracted by mixing 4 g ODE soil in 40 ml of 0.5 M hydrochloric acid (HCl) solution and shaking for 1 h followed by centrifugation. The HCl-extractable Fe(II) and Fe(III) (HCl-Fe(II) and HCl-Fe(III)) concentrations were determined using a modified ferrozine assay (Viollier et al. 2000).

Ammonium oxalate (AO) solution was used to estimate the concentrations of poorly crystalline Fe and Al minerals. A pre-treatment with 0.1 M HCl was used to remove Fe(II) before the AO extraction to avoid catalytic dissolution of crystalline Fe minerals in the presence of Fe(II) and oxalate at reducing conditions (Heiberg et al. 2012). Aliquots of 1 g ODE soil were mixed with 10 ml of 0.1 M HCl for 10 minutes on an end-to-end shaker, followed by centrifugation. Samples were then washed with 40 ml of $H_2O$, centrifuged, and subsequently used for AO extraction (40 mL) at pH 3.0 in the dark for 2 hours. The AO

extract was then filtered through 0.45 µm syringe filter before determination of Fe and Al concentrations (AO-Fe and AO-Al) with inductively coupled plasma optical emission spectrometry (ICP-OES) on three analytical replicates per sample (Perkin Elmer, Optima 5300 DV series, CA, USA).

Soil P fractions were estimated using a modified Hedley scheme after the pre-incubation and before the sorption test (Hedley

et al., 1982). Aliquots of 1 g ODE soil were sequentially extracted using 40 mL of 0.5 M sodium bicarbonate ($NaHCO_3$) and 0.1 M sodium hydroxide (NaOH) solution, each for 16 hours. The $NaHCO_3$ solution extracts inorganic and organic P that are weakly associated to soil particles and commonly assumed to be biologically available (Tiessen and Moir, 1993). The NaOH solution mobilizes P compounds that are more strongly bound to Fe and Al minerals than those extracted by $NaHCO_3$ solution, thus having intermediate availability (Tiessen and Moir, 1993). Total P concentration of both solutions was determined

following Murphy and Riley (1962) after autoclaving the solution with ammonium persulfate (($NH_4)_2S_2O_8$).

## 2.5 Phosphorus solubility experiments

We assessed the strength of P sorption by measuring the relative solubility of P for the slope and valley at the El Verde site. Overall, there were 96 samples (4 replicates × 2 extractants × 3 levels of P addition × 2 redox condition × 2 soil types).

Approximately 10 g ODE soil was weighed into pint-size glass jars and incubated either under ambient air or in an anaerobic glovebox for 14 days, following the method above. Samples then received 100 ml of 0.01 M $CaCl_2$ solution containing 0, 100, or 1000 mg P $kg^{-1}$ soil and 1 mL toluene. Jars were capped and shaken for 24 hours after which 5 mL subsamples of soil slurry were taken to estimate the amount of P remaining in solution and P sorbed by minerals using the methods described above. Microcosms then received 300 ml of either 0.667 M $NaHCO_3$ or 0.133 M NaOH solution to reach the final concentrations of

0.5 M $NaHCO_3$ and 0.1 M NaOH, respectively. The two extracting solutions were used because they have different P extraction efficiencies. Solutions were shaken every hour for three hours at both the beginning and the end of a 16-hour, overnight period prior to extraction. The anoxic treatment was conducted in the anoxic glove box with degassed solutions. Determination of



NaHCO$_3$ and NaOH total P concentrations (NaHCO$_3$-P$_t$ and NaOH-P$_t$, respectively) followed the previously described methods. The extracted P came from two potential sources: the P amendment (for the two treatments with P added) and native extractable P. We accounted for the native extractable soil P in the treatment without P addition. We also accounted for the amendment P remaining in solution in order to estimate the amount of *sorbed* P recovered during extraction. The recovered P

was then reported as a percentage of the sorbed P, or the relative P solubility.

## 2.6 Data analyses

Sorption isotherm data were modelled using the Langmuir equation as in Eq. (1):

$$S = \frac{aS_{max}C}{1+aC},\qquad(1)$$

in which $S$ is the amount of P sorbed during the batch experiment (mg P kg$^{-1}$ soil), $S_{max}$ corresponds to a predicted maximum sorption capacity (mg P kg$^{-1}$ soil), $C$ is the concentration of P remaining in solution (mg P L$^{-1}$), and $a$ represents a coefficient related to the bonding strength of P to soil minerals (L mg$^{-1}$ P). For each combination of soil type and redox treatment, all data points were used to fit the Langmuir equation using the nls function in R ver. 3.4.4 (R Core Team, 2018). This approach estimated the means and standard errors of all model parameters, including $S_{max}$. We also calculated a P sorption index (PSI,

L kg$^{-1}$ soil) (Bache and Williams, 1971) using the sorption isotherm results under all four P addition levels in Eq. (2):

$$PSI = \frac{S}{\log_{10} C},\qquad(2)$$

in which $S$ is the amount of P sorbed during the batch experiment (mg P kg$^{-1}$ soil), and $C$ corresponds to the concentration of P remaining in solution (mg P L$^{-1}$). High PSI values correspond to high P sorption capacity. Our discussion was focused on PSI values calculated at 1000 mg P kg$^{-1}$ because this rate of P addition was similar to the one used by Bache and Williams

(1971) (1,500 mg P kg$^{-1}$) and facilitated comparison with P sorption time curves measured at the same P addition rate. At P addition levels of 5000 and 10,000 mg P kg$^{-1}$, PSI values were also influenced by vivianite precipitation. Thus, we explored the relationships among PSI at 1000 mg P kg$^{-1}$, other P sorption indices, and soil Fe and Al concentrations.

Data from P sorption time curves were modelled using a power function as in Eq. (3):

$$100 - p = at^{-k},\qquad(3)$$

in which $p$ represents the percent of P sorbed (%), $t$ indicates time (h), $k$ corresponds to the rate of P sorption, and $a$ is a coefficient (h$^{-1}$). Higher values of $k$ indicate faster P sorption. Model fitting was also conducted using the nls function in R. Effects of redox manipulation on PSI values and P solubility were compared using student's T-tests in each soil at $\alpha = 0.05$ level. All analyses were conducted in R.






## 3 Results

### 3.1 Redox effects on P sorption

Sorption isotherms of all soils followed the Langmuir functions (Fig. 1). Estimated maximum sorption capacities ranged from $2526 \pm 667$ mg P kg$^{-1}$ to $8256 \pm 2517$ mg P kg$^{-1}$ (Table 2, mean $\pm$ S.E. unless otherwise noted), which was at least one order

of magnitude higher than total soil P concentrations (140-400 mg P kg$^{-1}$; Mage and Porder 2012). Effects of redox treatments on P remaining in solution and the P sorbed differed among levels of P addition. Under high P additions (5000-10,000 mg P kg$^{-1}$), P sorption (vertical axis Fig. 1) was generally greater under the anoxic treatment with lower concentrations of P remaining in solution relative to the oxic treatment (horizontal axis Fig. 1). Phosphorus sorption rates were high at low levels of addition (500 and 1000 mg P kg$^{-1}$, Fig. 1). When 1000 mg P kg$^{-1}$ was added, there was significantly lower P remaining in

solution under anoxic conditions at both slope soils ($P < 0.01$ at El Verde and $P < 0.001$ at Rio Icacos) and the valley soils at Rio Icacos ($P < 0.01$). Only the valley soils at El Verde showed significantly more P remaining in solution under anoxic conditions ($P < 0.01$). Similar trends were also observed at the lowest level of P addition (500 mg P kg$^{-1}$): there was significantly less P remaining in solution for the two slope soils under anoxic conditions ($P < 0.01$ at El Verde and $P < 0.001$ at Rio Icacos), and the opposite was true for the valley soils at El Verde ($P < 0.01$). However, the valley soils at Rio Icacos

showed no significant differences in solution P remaining between the two redox treatments.

The PSI values offered another way to examine the redox treatments on P sorption. Under high P additions (5000-10,000 mg P kg$^{-1}$), anoxic conditions led to greater or similar PSI values as oxic conditions (Table S1). When 500 mg P kg$^{-1}$ was added, PSI values became negative in two valley soils under oxic conditions because their average P concentrations remaining in

solution were lower than 1 mg L$^{-1}$, resulting in negative logarithms. Under P addition of 1000 mg P kg$^{-1}$, anoxic conditions led to lower PSI values than oxic conditions in the El Verde valley soil ($P < 0.05$, Fig. 2), while redox incubations did not significantly affect PSI in the valley soil from Rio Icacos. In contrast, anoxic conditions increased PSI values relative to the oxic treatment in the two slope soils (both $P < 0.01$). The valley soils had $116 \pm 25\%$ and $64 \pm 14\%$ higher PSI values than the slope soils at El Verde and Rio Icacos, respectively (both $P < 0.05$). Averaging between topographic positions, differences in

PSI between the two sites were relatively small (El Verde vs. Rio Icacos: $720 \pm 89$ L kg$^{-1}$ soil vs. $890 \pm 7$ L kg$^{-1}$ soil, $P < 0.05$) compared to the effects of redox conditions and topographic positions. For the remainder of the text, we will refer to PSI values as those calculated under P addition of 1000 mg P kg$^{-1}$ unless noted otherwise.

Rapid P sorption was observed in all soils, as at least 60% of the added P had disappeared from solution within the first six

hours of sorption experiment (Fig. 3). In the two slope soils, P sorption occurred more rapidly after anoxic incubation than after oxic incubation, as indicated by the lower concentrations of P remaining in solution under anoxic conditions (all $P < 0.05$ after 12 hours). In valley soils from El Verde, however, P sorption occurred more rapidly after oxic incubation, as $2.6 \pm 0.4\%$



vs. 30.3 ± 3.3% of the added P remained in solution after the first six hours after oxic and anoxic incubation, respectively ($P$ < 0.001). In valley soils from Rio Icacos, more P was sorbed in soils from the oxic treatment than those from anoxic treatment during the first six hours (all $P$ < 0.05), while afterwards no effects of redox treatment were found. The rate of P sorption ($k$) was strongly correlated with PSI values ($r$ = 0.86, $P$ < 0.01, Table 2).

### 3.2 Redox effects on other soil characteristics

We measured soil Fe and Al concentrations to determine relationships with P sorption indices. Soil HCl-Fe(III) concentrations were significantly higher at El Verde than at Rio Icacos ($P$ < 0.001, Table 2). Anoxic conditions decreased soil HCl-Fe(III) concentrations relative to oxic conditions in the slope ($P$ < 0.01) and valley ($P$ < 0.001) soil at El Verde. However, soil HCl-

Fe(III) concentrations did not respond significantly to redox treatments in two soils from Rio Icacos. In all four types of soils, the anoxic treatment had significantly higher soil HCl-Fe(II) concentrations relative to the oxic treatment (up to two orders of magnitude greater), providing evidence that soils experienced reducing conditions (Table 2). This effect was particularly strong in the slope soil at El Verde and at Rio Icacos (both $P$ < 0.001). Among all samples, soil HCl-Fe(III) concentrations were weakly positively correlated with PSI values ($r$ = 0.43, $P$ < 0.05), while HCl-Fe(II) values were not. The correlation between

HCl-Fe(III) and PSI was strongest at El Verde (Fig. S1; $r$ = 0.89, $P$ < 0.001).

Soils at El Verde had higher AO-Fe concentrations ($P$ < 0.001, Table 2) and lower AO-Al concentrations ($P$ < 0.001) than those at Rio Icacos (Table 2). Within the El Verde samples, the valley soil had higher AO-Fe concentrations ($P$ < 0.01) and lower AO-Al concentrations ($P$ < 0.001) than the slope soil. Within the Rio Icacos site, the valley soil had nearly doubled the

AO-Al concentrations compared to the slope soil ($P$ < 0.001), while the two topographic zones had similar levels of AO-Fe. Anoxic conditions decreased AO-Fe concentrations in the valley soils from Rio Icacos ($P$ < 0.001), and increased concentrations of AO-Al in the valley soil at El Verde ($P$ < 0.05) (Table 2). Among all samples, concentrations of AO-Al were weakly positively correlated with PSI values ($r$ = 0.39, $P$ <0.05), but AO-Fe were not. A positive correlation between AO-Al concentrations and PSI was found at Rio Icacos (Fig. S1; $r$ = 0.74, $P$ = 0.002), while their correlation was negative at El Verde

($r$ = -0.71, $P$ = 0.002).

### 3.3 Phosphorus solubility

We explored the extractability of P following P additions at the El Verde site. The average relative solubility of P was much lower in NaHCO$_3$ solution than in NaOH solution (20.5 ± 1.3% vs. 69.7 ± 2.1%; $P$ < 0.001, Fig. 4). Phosphorus additions of

100 mg P kg$^{-1}$ yielded higher *relative* P solubility than 1000 mg P kg$^{-1}$ additions, in both NaHCO$_3$ ($P$ < 0.001) and NaOH solutions ($P$ < 0.001), as extractants might have reached their limits of dissolving P under higher P addition level. Anoxic





conditions generally increased P solubility in NaHCO$_3$ solution, except in the slope soil with P addition of 100 mg P kg$^{-1}$. Phosphorus solubility in NaOH solution decreased in the anoxic treatment in the valley soil with the lower level of P addition only ($P < 0.001$). Note that anoxic conditions increased PSI values in both soils under P additions of 1000 mg P kg$^{-1}$ (Fig. S2).

## 4 Discussion

### 4.1 Anoxic conditions maintained high P sorption

Contrary to our first hypothesis, anoxic conditions led to similar or greater rates of P sorption as oxic conditions in all but one treatment (P sorption time curve for El Verde valley). This suggests that soils remain strong P sinks even under reducing conditions. Under both oxic and anoxic conditions, estimated maximum P sorption capacities were much higher than the range

of total soil P concentrations (Mage and Porder, 2012), highlighting the significant potential of these soils for retaining P. Sorption of P occurred very rapidly under both redox conditions, with over 60% of the added P removed from solution within six hours in all soils. Results indicate that low redox events are unlikely to induce significant P release to the soil solution in these soils. High P sorption potential is very likely responsible for the extremely low P concentrations of stream water in local watersheds (McDowell, 1998; McDowell and Liptzin, 2014). Our results also suggest that new P entering the ecosystem via

atmospheric sources such as dust or smoke (Pett-Ridge, 2009) would likely be rapidly sorbed by soil minerals.

Three mechanisms are likely responsible for the high P sorption capacities under anoxic conditions. First, mixed Fe(III)-Fe(II) or Fe(II) minerals formed during Fe reduction may have a high sorption capacity. These minerals can feature a more amorphous structure compared to Fe(III) phases and thus have a higher surface area available for P sorption (Patrick and Khalid, 1974;

Holford and Patrick, 1979; Borch and Fendorf, 2007). Soils from the study site have very large and diverse populations of microbial Fe reducers that facilitate rapid Fe reduction (Dubinsky et al., 2010), contributing to the formation and maintenance of high amorphous Fe minerals. Second, Fe reduction is known to increase soil pH (Lindsay, 1979) that consequently increases the degree of hydroxylation and surface area of Al and organo-Al complexes (Haynes, 1982). These changes in Al speciation have commonly been used to explain the increased P sorption capacities under liming (Haynes and Swift, 1989; Gustafsson et

al., 2012). Finally, formation of Fe(II)-P minerals, such as vivianite, can also contribute to high P retention (Heiberg et al., 2012; Walpersdorf et al., 2013). In our experiment, the soil slurry appeared to be supersaturated with respect to vivianite precipitation as calculated by Visual MINTEQ (Gustafsson, 2011), especially under high P loadings (5000 and 10,000 mg P kg$^{-1}$, Table S2). However, the effect of vivianite formation was likely to be small under low P loadings (e.g., 1000 mg P kg$^{-1}$), as its precipitation kinetics is slow and depends on P concentration (Borch and Fendorf, 2007; Heiberg et al., 2012).




## 4.2 Phosphorus sorption and Fe and Al minerals

Our results identified amorphous Fe(III) and Al minerals as the best predictors of soil PSI values, but not Fe(II) minerals. These results suggest that soil Fe(II) minerals alone were insufficient to explain P sorption capacity compared to Fe(III) or Al minerals, consistent with previous studies (Sallade and Sims, 1997; Quintero et al., 2007; Rakotoson et al., 2014). However,

soil Fe(III) and Al minerals were less responsive to redox manipulation compared to Fe(II) minerals. Persistence of amorphous minerals, as well as crystalline Fe and Al minerals, are all likely contributors to the high P sorption capacity under both redox conditions (Gérard, 2016; McGechan and Lewis, 2002). We found that anoxic conditions did not affect HCl-Fe(III) concentrations in soils from Rio Icacos, and that changes in HCl-Fe(III) and AO-Fe were smaller in magnitude compared to the increases in HCl-Fe(II) across all soil types. Together, this indicates that Fe reduction mobilized crystalline or poorly

crystalline Fe that was not soluble in HCl or AO solution. Overall, concentrations of soil Fe(III) and Al minerals influenced soil P sorption behavior across soil types, and their persistence contributed to high P sorption under both redox environments.

Patterns in soil Fe and Al concentrations helped to explain the variability of soil P sorption behavior across the two topographic zones. Valley soils, which are characterized by frequent high magnitude redox fluctuations and low redox events (Silver et al.,

1999), had higher concentrations of HCl-Fe(III) than slope soils at both sites, and had similar or higher levels of AO-Fe. The valley soil at Rio Icacos had twice as much AO-Al as the slope soil. The higher levels of amorphous Fe and Al minerals in valley soils likely contributed to higher PSI values relative to slopes in both locations. The differences in amorphous Fe and Al concentrations likely resulted from the long-term changes in redox history and soil transport along the catena (Hall and Silver, 2015; Mage and Porder, 2012). Water and organic matter transport leads to higher soil moisture content in valleys than

in slope soils and helps to create more frequent and intensive reducing events (Silver et al., 1999). These conditions facilitate the formation and persistence of amorphous Fe and Al minerals.

Differences in Fe and Al minerals also helped to explain the patterns of P sorption across two sites. Soils at El Verde were enriched in HCl-Fe(III) and AO-Fe but depleted in AO-Al compared to soils at Rio Icacos, and the positive correlation between

HCl-Fe(III) and PSI was only significant at El Verde. Soil AO-Al concentrations were positively correlated with PSI at Rio Icacos, but had a negative correlation at El Verde. These results suggest that Fe minerals may play a primary role in sorbing P in the volcanoclastic soils, while Al minerals were more important to P sorption in the dioritic soils. Together our results show that redox history and parent material influenced the patterns of soil P sorption across topographic zones and study site, respectively, at long timescales.


In the valley soil from El Verde, the effects of redox manipulation on P sorption differed greatly among levels of P addition, a pattern not observed in the other three sampling sites. High concentrations of Fe(II), derived from the low-redox legacy of the soil, and soluble P under high P loadings (5000 and 10,000 mg P kg$^{-1}$) represented favorable conditions for vivianite





formation, which likely contributed to the higher P sorption capacity under anoxic conditions in this soil. Vivianite formation did not appear to play a major role in P sorption at a lower P loading (1000 mg P kg$^{-1}$), as the anoxic treatment decreased the PSI value and P sorption rate in this soil. The strong decline in amorphous Fe(III) concentrations as a result of Fe reduction was potentially responsible for the reduced PSI and P sorption rate under anoxic conditions in these soils. The specific effects

of different P-loadings under redox manipulation also led to the lack of significant correlation between $S_{max}$ and PSI values, because $S_{max}$ was mostly driven by data from high P loadings, while PSI was calculated at the P loading of 1000 mg P kg$^{-1}$.

The PSI values of the valley soil at El Verde showed different responses to anoxic incubation in two separate trials. Soil moisture content measured in the P solubility experiment was significantly lower than its mean value calculated from

continuous field observations (O'Connell et al., 2018) and lower than that measured in the initial trial, likely due to natural background variability in rainfall. The PSI value and its response to redox manipulation resembled those of the slope soil. It is possible that the dry period decreased the reactive surface areas of soil minerals potentially by oxidizing reduced species and increasing the crystallinity of secondary Fe and Al minerals. These results suggest that soil P dynamics could be highly sensitive to changes in environmental conditions in tropical forests (O'Connell et al., 2018).

### 4.3 Implications for P solubility and bioavailability

Our results showed that, regardless of redox conditions, significantly more P was soluble in NaOH than NaHCO$_3$ and that the relative P solubility was high (> 50% of sorbed P) in NaOH for both the 100 and 1000 mg P kg$^{-1}$ additions. The NaOH extraction is stronger than the NaHCO$_3$ solution, and together they are thought to represent a continuum of P bound to Fe and

Al minerals (Tiessen and Moir 1993). Anoxic conditions generally increased the solubility of the sorbed P in NaHCO$_3$ solution, indicating that the strength of P sorption was weaker when soils had more anaerobic microsites. The change in P sorption likely reflected the lower binding strengths of reduced Fe minerals to P than their oxidized forms (Zhang et al., 2003; Holford and Patrick, 1979). Past research has reported increased P solubility in response to Fe reduction using soils from the tropics (Peretyazhko and Sposito, 2005; Liptzin and Silver, 2009; Chacón et al., 2006; Maranguit et al., 2017; Lin et al., 2018). Our

results suggest that Fe-redox dynamics increase P solubility via decreasing the strength, rather than the capacity of soil P sorption.

Our results have important implications for understanding P bioavailability in tropical forest soils. These results suggest that reducing events can potentially increase P bioavailability by decreasing the P sorption strength of minerals, even though P

sorption capacity remained high. An interesting question for future studies is whether plant roots and microbes can take advantage of the increased P solubility during reducing events. Anaerobic conditions can be stressful for plants and microbes; however, studies have reported similar soil respiration rates under aerobic and anaerobic conditions in humid tropical forest

soils (DeAngelis et al., 2010; Pett-Ridge, 2005). Soil microbes appear to be well-adapted to dynamic redox conditions, at least at the scales of days to weeks. Thus, it is possible that microbes can benefit from the increased P solubility under anoxic conditions. At a watershed scale, topography and parent material are important controls of soil sorption behavior and P bioavailability. Although P sorption capacity was high in soils with high Fe and Al concentrations, these soils also responded

more strongly to reducing events. Thus, redox dynamic may be particularly important in these soils to facilitate biological uptake and ecosystem retention of P.

## 5 Conclusions

We found that minerals can retain high P sorption capacity during reducing events in highly weathered tropical forest soils. The high P sorption capacity is expected to contribute to low P concentrations in soil solution and limit the potential for P

leaching. Due to the high P sorption capacity, current and future increases in precipitation associated with climate change are unlikely to drastically alter P leaching in these environments. Reducing events also decreased the strength of P sorption and potentially increased P bioavailability. Thus, episodic reducing events could serve as 'hot moments' for plants and microbes to acquire soil P that would otherwise be tightly bound to minerals. As a result of altered rainfall regimes, more frequent or intensive redox oscillation could increase P bioavailability, if it does not impose a strong $O_2$ limitation on primary productivity

or decomposition.

**Author contributions**: YL, AG, and WLS conceived the study; YL, AG, and CSO performed the research; YL led the manuscript development and data analysis; all authors contributed to writing.

**Competing interests**: The authors declare that they have no conflict of interest.

**Acknowledgements**: We thank Summer Ahmed, Heather Dang, Jordan Stark, Omar Gutiérrez del Arroyo, Sarah Stankavich, and Gisela Gonzalez for their support in the laboratory and in the field. This study benefited from discussion with Aaron Thompson, Chunmei Chen, Steven Hall, and Tyler Anthony. This work was supported by grants to WLS from the National

Science Foundation (DEB-1457805, Luquillo CZO EAR-1331841, and LTER DEB-0620910), as well as the Department of Energy (TES-DE-FOA-0000749). WLS was also supported by the USDA National Institute of Food and Agriculture, McIntire Stennis project CA-B-ECO-7673-MS. Data from this study will be made available via the Luquillo CZO and Hydroshare (http://www.hydroshare.org/) upon the acceptance for publication.

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





Table 1. Initial soil characteristics ($n = 4$).

| Site | Position | Gravimetric soil moisture (%) | pH | Total soil C (%) | Total soil N (%) | |
|------|----------|-------------------------------|-----|------------------|------------------|----|
| El Verde 1 | Slope | 95.7 ± 2.4 | 5.27 ± 0.02 | 7.06 ± 0.37 | 0.386 ± 0.007 | 5 |
| | Valley | 124.5 ±1.3 | 5.70 ± 0.06 | 5.91 ± 0.16 | 0.307 ± 0.004 | |
| Rio Icacos | Slope | 71.6 ± 0.4 | 4.72 ± 0.01 | 3.90 ± 0.06 | 0.196 ± 0.002 | |
| | Valley | 103.7 ± 0.3 | 5.26 ± 0.06 | 4.28 ± 0.11 | 0.237 ± 0.002 | |
| El Verde 2 | Slope | 85.1 ± 2.4 | n.d. | n.d. | n.d. | |
| P solubility | Valley | 78.5 ± 0.4 | n.d. | n.d. | n.d. | 10 |

*Means and S.E. are shown. n.d., not determined*



Table 2. Maximum P sorption capacities ($S_{max}$), rates of P sorption ($k$), and soil Fe and Al fractions in response to redox treatments and their correlations with phosphorus sorption index (PSI) calculated from sorption data under 1000 mg P kg$^{-1}$ ($n$ = 4)

| Site | Position | Treat-ment | $S_{max}$ (mg P kg$^{-1}$) | $k$ | HCl-Fe(II) (g Fe kg$^{-1}$) | HCl-Fe(III) (g Fe kg$^{-1}$) | AO-Fe (g Fe kg$^{-1}$) | AO-Al (g Al kg$^{-1}$) |
|---|---|---|---|---|---|---|---|---|
| El Verde | Slope | Anoxic | 7451 ± 1171 | 0.275 ± 0.020 | 3.37 ± 0.12a | 0.86 ± 0.09b | 0.92 ± 0.06a | 0.29 ± 0.03a |
| | | Oxic | 5453 ± 566 | 0.244 ± 0.016 | 0.038 ± 0.003b | 1.55 ± 0.12a | 0.75 ± 0.09a | 0.30 ± 0.03a |
| | Valley | Anoxic | 7142 ± 558 | 0.288 ± 0.027 | 13.72 ± 0.22a | 1.85 ± 0.08b | 1.37 ± 0.03a | 0.20 ± 0.004a |
| | | Oxic | 7282 ± 1207 | 0.486 ± 0.049 | 4.06 ± 0.69b | 5.63 ± 0.55a | 1.33 ± 0.23a | 0.17 ± 0.01b |
| Rio Icacos | Slope | Anoxic | 6291 ± 1469 | 0.303 ± 0.015 | 3.36 ± 0.17a | 0.67 ± 0.13a | 0.29 ± 0.01a | 1.00 ± 0.04a |
| | | Oxic | 2526 ± 667 | 0.194 ± 0.013 | 0.060 ± 0.002b | 0.80 ± 0.05a | 0.31 ± 0.03a | 0.98 ± 0.07a |
| | Valley | Anoxic | 8256 ± 2519 | 0.365 ± 0.022 | 3.39 ± 0.30a | 0.90 ± 0.06a | 0.29 ± 0.004b | 2.01 ± 0.05a |
| | | Oxic | 3290 ± 250 | 0.353 ± 0.025 | 0.193 ± 0.124b | 0.92 ± 0.10a | 0.34 ± 0.01a | 1.89 ± 0.05a |
| Correlation with PSI ($n$ =32) | | $r$ | 0.366‡ | **0.863‡** | 0.092 | **0.434** | 0.001 | **0.395** |
| | | $P$ | 0.372‡ | **0.006‡** | 0.616 | **0.013** | 0.998 | **0.028** |

*Means and S.E. are shown. Different letters indicate significant effects of redox treatment in each combination of site and*

5  *topographic position (T-tests). Significant correlation coefficient (r) and P values are in bold. ‡, n = 8.*



**Figure 1. Effects of anoxic vs. oxic pre-incubation on the P sorption isotherms of slope (left panel) and valley (right panel) soils from El Verde (upper panel) and Rio Icacos (lower panel). Means and standard errors of means are shown. Inserts are the zoomed-in versions of the lower left portion of the respective plots.**



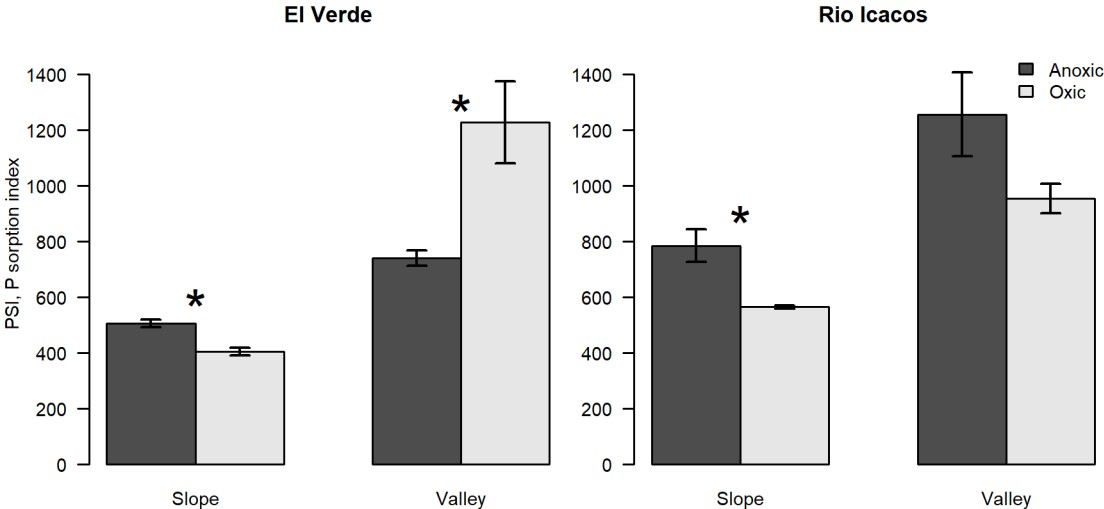

**Figure 2. Effects of anoxic vs. oxic pre-incubation on the P sorption index (PSI) of slope and valley soils from El Verde and Rio Icacos. The PSI values were calculated from sorption isotherm data under 1000 mg P kg⁻¹, which better represented P sorption behavior under low levels of P addition than the estimated maximum P sorption capacity. Means and standard errors of means are shown. * indicates significant difference of PSI for the respective combination of site and topographic position.**





**Figure 3. Effects of anoxic vs. oxic pre-incubation on the P sorption time curves of slope (left panel) and valley (right panel) soils from El Verde (upper panel) and Rio Icacos (lower panel). Phosphorus was added at the level of 1000 mg P kg$^{-1}$. Means and standard errors of means are shown.**



**Figure 4.** Effects of anoxic vs. oxic pre-incubation on the relative solubility of the sorbed P in NaHCO$_3$ and NaOH solution under P addition of 100 mg P kg$^{-1}$ (P100, left panel) and 1000 mg P kg$^{-1}$ (P1000, right panel). Soils were collected from slope and valley positions at El Verde. Means and standard errors of means are shown. * indicates significant difference of PSI for the respective combination of site, topographic position, and level of P addition.