# Peer review of "Anoxic conditions maintained high phosphorus sorption in humid tropical forest soils"

_Biogeosciences, 2019_

## Referee Comment (RC1) · Eric Roy (Referee) · 25 Mar 2019

General comments: This manuscript by Lin et al. examines the influence of anoxic conditions on phosphorus sorption in highly weathered, acidic soils in humid tropical forests. It is well known that phosphorus solubility and bioavailability in soils and sediments can be impacted by fluctuations in redox conditions that lead to iron reduction and pH changes. However, to my knowledge (and based on the literature reviewed by Lin et al.), this topic has not been well studied in tropical forest soils (or tropical soils used for agriculture) that are rich in Fe- and Al-minerals and characterized by large P sorption capacity. Therefore, this is a welcome study and the authors have done a nice job generating results that increase understanding of this topic. They perform their experiments using humid tropical forest soils (0-15 cm depth) from two sites in Puerto

[Figure]

Rico featuring different parent materials and soil characteristics. Tests include P sorption isotherms, P sorption time curves, soil Fe and Al analyses and P fractions, and a P solubility experiment. The methods used are appropriate for the study questions. I support the publication of this manuscript in Biogeosciences once my comments below have been taken into account.

Specific comments:

1) Re: the choice of P concentrations for the sorption isotherm experiments: a. Can the authors provide more justification for their decision to omit the lower concentrations (i.e., 10 and 100 mg P kg-1 soil)? Typically, sorption isotherm experiments have included a 0 mg P kg-1 soil treatment, as well as lower concentrations where most P is sorbed (e.g., Graetz and Nair 2000). Would including these results influence the Langmuir models in this study? b. The range of concentrations used (500 – 10,000 mg P kg-1 soil) is very high. It should be noted that estimates for Smax can be dependent on the concentrations used in the P sorption isotherm experiment, particularly when testing a material with high P sorption capacity. For example, Drizo et al. (2002) found that the Smax for a steel slag material tested for use in constructed wetlands varied from 0.31 to 3.93 g P kg-1 material depending on the range of P concentrations used. Here, the authors report maximum P sorption capacities ranging from 2526 $\pm$ 667 mg P kg-1 to 8256 $\pm$ 2517 mg P kg-1 soil. These are high compared to other studies of P-fixing soils that used lower, more conventional ranges in P concentrations (e.g., Brazilian forest soils tested in Roy et al. 2017 using the range of 0 - 1,500 mg P kg-1 soil exhibited Smax up to 1167 mg P kg-1 soil). I think the authors should provide some discussion of whether their Smax results are method-dependent, and how their results compare to other studies, including those that used more conventional, lower ranges of P concentrations. This discussion can add nuance to their conclusion that Smax was "at least one of magnitude above total soil P concentrations."

2) The authors state that "sorption isotherms of all soils followed Langmuir functions", however, no model fit diagnostics appear in the manuscript. Can the authors please

provide quantitative evidence to justify this statement? The standard errors for Smax presented seem somewhat high to me and it would be helpful to know how well the Langmuir model fit the data. Model efficiency is one option (Bolster and Hornberger 2007).

3) Page 6, Line 20 – Can the authors please explain here how they determined that vivianite precipitation was occurring? There is some discussion on subsequent pages that I suggest be moved to the first mention of vivianite precipitation. Furthermore, the calculations made using Visual MINTEQ should be described somewhere in the Methods.

4) Page 8, Line 10 – The evidence that soils experienced reducing conditions mentioned here should probably come earlier in the paper given its importance for all results presented.

References

Bolster, C. H., & Hornberger, G. M. (2007). On the use of linearized Langmuir equations. Soil Science Society of America Journal, 71(6), 1796-1806.

Drizo, A., Comeau, Y., Forget, C., & Chapuis, R. P. (2002). Phosphorus saturation potential: a parameter for estimating the longevity of constructed wetland systems. Environmental science & technology, 36(21), 4642-4648.

Graetz, D. A., & Nair, V. D. (2000). Phosphorus sorption isotherm determination. In Methods of phosphorus analysis for soils, sediments, residuals, and waters (Vol. 369, pp. 35-38). North Carolina State University Raleigh.

Roy, E. D., Willig, E., Richards, P. D., Martinelli, L. A., Vazquez, F. F., Pegorini, L., ... & Porder, S. (2017). Soil phosphorus sorption capacity after three decades of intensive fertilization in Mato Grosso, Brazil. Agriculture, ecosystems & environment, 249, 206-214.

---

## Referee Comment (RC2) · Anonymous Referee #2 · 24 Jun 2019

General Comments

The manuscript "Anoxic conditions maintained high phosphorus sorption in humid tropical forest soils" describes an experiment using tropical soils from the Luquillo experimental station to assess the effect of anoxic conditions on the adsorption capacity, kinetics and P sorption strength in those soils. Two sets of soils, two topographical units (slope and valley) and two oxic conditions (anoxic and oxic) were used. The results are of great importance for the biogeochemistry community because: (1) few studies on the influence of anoxic conditions to the P sorption capacity/strength in tropical soils are published, (2) because it contradicts the stablished assumption that reducing conditions increase P availability. While I have no specific grammar corrections, the fluidity of the text should be revisited. Moreover, inconsistencies in the methods, results and

discussion sections of the manuscript are visible. I would recommend the publication of the study, after the following major revisions.

Major comments

Although the authors have executed a very interesting work, lack of information on the sample prior to analysis and subsequent analysis during the discussion are not well documented. The authors justify their chosen topographical units as inherent more oxic and less oxic sites yet, no information on the P content, FeOA, AlOA, FeHCl and AlHCl prior incubation is available in Table 1, while the reader is referred to several citations, the addition of this data to Table 1 would be very beneficial. Moreover, while the authors aim to compare the effect on anoxic conditions on the soil capacity to adsorb P no information on the soil specific surface area (SSA) before and after the experiment is available. Moreover, I would recommend the use of SSA for isotherm to analyze the P loading on the minerals (g P/m2) while adsorption occurred.

We also question the use of just 4 points in each isotherm, without lower values or zero, the lack of fit statistics (r2, among others) and general lack of detail. Later the authors also rely in the lower values of PSI to analyze and compare their sorption curves, while they also refer to the PSI concentration as rate in page 6 line 19. The authors also mention several times the precipitiation of viviatine, while no evidence more than the mention of a MINTEQ simulation to the reader. Altough the information in in the supplementary section the reader is never refered to it. The details of this MINTEQ simulation are also omitted in the Methods section.

At the results, the authors describe the P concentrations added as rates, this is extremely confusing as rates refer to a quantity over time. Which would be the kinetic data. The authors also continue to discuss data from figure 1 (adsorption isotherms) while comparing p-values for the different concentrations of the isotherm while no table or figure is mentioned. The data that this refers is supplementary table 1, where the author refers to the rates as levels, yet in the paragraph no mention to this supplementary

table is made.

In the discussion, the authors disregarded their kinetic and solubility data and support their discussion on the PSI and its correlation with AlOA and FeOA. The author does not seek to discuss the nuances a faster sorption rate at their simulated anoxic condition in the slope sites. On the other hand, the authors never discuss what minerals/solid phases could be the ones extracted by their HCl(FeII) and HCl(FeII) and how is this related to the higher P soption and rates in some soils. They base their conclusion on their solubility analysis yet this information is never related to the previous analysis. I would recommend the authors to discussing their results in comparison with the study "Sorption isotherms and kinetics of sediment phosphorus in a tropical reservoir" by Adhityan Appan, and Hong Wang; which is very similar to theirs.

Minor revisions

Page 12 line 9: change The to their

Page 12 line 10 change "due to the" to "do to the tropical soils'"

---

## Author Comment (AC1) · 12 Aug 2019

Reviewer #1 (Dr. Eric Roy)

General comments: This manuscript by Lin et al. examines the influence of anoxic conditions on phosphorus sorption in highly weathered, acidic soils in humid tropical forests. It is well known that phosphorus solubility and bioavailability in soils and sediments can be impacted by fluctuations in redox conditions that lead to iron reduction and pH changes. However, to my knowledge (and based on the literature reviewed by Lin et al.), this topic has not been well studied in tropical forest soils (or tropical soils used for agriculture) that are rich in Fe- and Al-minerals and characterized by large P sorption capacity. Therefore, this is a welcome study and the authors have done a nice job generating results that increase understanding of this topic. They perform their experiments using humid tropical forest soils (0-15 cm depth) from two sites in Puerto Rico featuring different parent materials and soil characteristics. Tests include P sorption isotherms, P sorption time curves, soil Fe and Al analyses and P fractions, and a P solubility experiment. The methods used are appropriate for the study questions. I support the publication of this manuscript in Biogeosciences once my comments below have been taken into account.

**We appreciate Dr. Roy's interest in our work.**

Specific comments: 1) Re: the choice of P concentrations for the sorption isotherm experiments: a. Can the authors provide more justification for their decision to omit the lower concentrations (i.e., 10 and 100 mg P kg-1 soil)? Typically, sorption isotherm experiments have included a 0 mg P kg-1 soil treatment, as well as lower concentrations where most P is sorbed (e.g., Graetz and Nair 2000). Would including these results influence the Langmuir models in this study?

**We did not include a P-blank treatment because there was no detectable water-extractable P (or anion exchange resin P, which is higher in concentration) in these soils (McGroddy and Silver 2000 Biotropica). Please see the next section for our response to the comment that our P sorption experiments did not include low P concentration.**

b. The range of concentrations used (500 – 10,000 mg P kg-1 soil) is very high. It should be noted that estimates for Smax can be dependent on the concentrations used in the P sorption isotherm experiment, particularly when testing a material with high P sorption capacity. For example, Drizo et al. (2002) found that the Smax for a steel slag material tested for use in constructed wetlands varied from 0.31 to 3.93 g P kg-1 material depending on the range of P concentrations used. Here, the authors report maximum P sorption capacities ranging from 2526 ± 667 mg P kg-1 to 8256 ± 2517 mg P kg-1 soil. These are high compared to other studies of P-fixing soils that used lower, more conventional ranges in P concentrations (e.g., Brazilian forest soils tested in Roy et al. 2017 using the range of 0 - 1,500 mg P kg-1 soil exhibited Smax up to 1167 mg P kg-1 soil). I think the authors should provide some discussion of whether their Smax results are method-dependent, and how their results compare to other studies, including those that used more conventional, lower ranges of P concentrations. This discussion can add nuance to their conclusion that Smax was "at least one of magnitude above total soil P concentrations."

[Figure]

*As mentioned in the manuscript, our preliminary trials included low P concentrations (10, 100, 500, and 1000 mg P kg-1). As shown in the figure above, the two Icacos soils showed P sorption capacity (Smax) extremely close to or higher than the maximum P addition (1000 mg P kg-1). Furthermore, average P concentrations in solution were 0.01 and 0.04 mg P L-1 in samples receiving 10 and 50 mg P kg-1, respectively. Many samples showed no detectable P in solution. Thus, we needed to use higher P concentrations to discern actual sorption potentials. We included the above justification in the methods (Page 4 Line 15-20) and added the figure to the supplement (Figure S1).*

*We agree with Dr. Roy that concentrations of P would influence the estimation of sorption capacity. We have added the following text in the discussion: "Our estimates of $S_{max}$ were relatively high relative to other humid tropical soils. For example, de Campos et al. (2016) applied up to 8000 mg P kg-1 to a set of strongly weathered Brazilian forest soils and reported a wide range of $S_{max}$ (61-5460 mg P kg-1). Compared to our preliminary trials with maximum P addition of 1000 mg P kg-1 (Fig. S1), $S_{max}$ were higher when more P was added (5000 and 1,0000 mg P kg-1; Table 2), suggesting that $S_{max}$ can be influenced by the concentrations of P used in the experiment. However, even when the maximum P addition level was similar (up to 1500 mg P kg-1 soil), $S_{max}$ from the preliminary trials were also high compared to other strongly weathered soils, including Brazilian forest soils (295-1167 mg P kg-1, Roy et al., 2017) and Thai upland soils (47-1250 mg P kg-1, Wisawapipat et al., 2009).".*

2) The authors state that "sorption isotherms of all soils followed Langmuir functions", however, no model fit diagnostics appear in the manuscript. Can the authors please provide quantitative evidence to justify this statement? The standard errors for Smax presented seem somewhat high to me and it would be helpful to know how well the Langmuir model fit the data. Model efficiency is one option (Bolster and Hornberger 2007).

*We adopted the Bolster and Hornberger's method as recommended by the reviewer and updated the estimates of Smax and standard errors. Model efficiency ranged between 0.781 to 0.967, indicative of good model fit (Page 7 Line 23-24). Results of model efficiency were included in the supplement (Table S1).*

3) Page 6, Line 20 – Can the authors please explain here how they determined that vivianite precipitation was occurring? There is some discussion on subsequent pages that I suggest be moved to the first mention of vivianite precipitation. Furthermore, the calculations made using Visual MINTEQ should be described somewhere in the Methods.

> *We suspected that vivianite precipitation would contribute to P sorption in anoxic treatments, as the concentrations of P and Fe(II) were high in these samples. We moved some of the discussion from later pages to the results where we first mentioned vivianite precipitation. We also added information on the Visual MINTEQ modeling to the methods (Page 6 Line 13-22).*

4) Page 8, Line 10 – The evidence that soils experienced reducing conditions mentioned here should probably come earlier in the paper given its importance for all results presented.

> *We agreed with this comment and reorganized the results to emphasize the effects of redox treatments (Page 7, Lines 5-21).*

References Bolster, C. H., & Hornberger, G. M. (2007). On the use of linearized Langmuir equations. Soil Science Society of America Journal, 71(6), 1796-1806. Drizo, A., Comeau, Y., Forget, C., & Chapuis, R. P. (2002). Phosphorus saturation potential: a parameter for estimating the longevity of constructed wetland systems. Environmental science & technology, 36(21), 4642-4648. Graetz, D. A., & Nair, V. D. (2000). Phosphorus sorption isotherm determination. In Methods of phosphorus analysis for soils, sediments, residuals, and waters (Vol. 369, pp. 35-38). North Carolina State University Raleigh. Roy, E. D., Willig, E., Richards, P. D., Martinelli, L. A., Vazquez, F. F., Pegorini, L., ... & Porder, S. (2017). Soil phosphorus sorption capacity after three decades of intensive fertilization in Mato Grosso, Brazil. Agriculture, ecosystems & environment, 249, 206- 214.

---

## Author Comment (AC2) · 12 Aug 2019

Reviewer #2:

General Comments The manuscript "Anoxic conditions maintained high phosphorus sorption in humid tropical forest soils" describes an experiment using tropical soils from the Luquillo experimental station to assess the effect of anoxic conditions on the adsorption capacity, kinetics and P sorption strength in those soils. Two sets of soils, two topographical units (slope and valley) and two oxic conditions (anoxic and oxic) were used. The results are of great importance for the biogeochemistry community because: (1) few studies on the influence of anoxic conditions to the P sorption capacity/strength in tropical soils are published, (2) because it contradicts the stablished assumption that reducing conditions increase P availability. While I have no specific grammar corrections, the fluidity of the text should be revisited. Moreover, inconsistencies in the methods, results and discussion sections of the manuscript are visible. I would recommend the publication of the study, after the following major revisions.

> ***We appreciate reviewer's comments. We improved the fluidity of the text during the revision. We also fixed the inconsistencies identified later by the reviewer, including the lack of background soil chemistry data (Table 1) and the inappropriate use of the term 'rate'. We made sure that the rate of P sorption was only used when discussing coefficient k of the P sorption curve (e.g., Page 9 Line 14).***

Major comments Although the authors have executed a very interesting work, lack of information on the sample prior to analysis and subsequent analysis during the discussion are not well documented. The authors justify their chosen topographical units as inherent more oxic and less oxic sites yet, no information on the P content, FeOA, AlOA, FeHCl and AlHCl prior incubation is available in Table 1, while the reader is referred to several citations, the addition of this data to Table 1 would be very beneficial. Moreover, while the authors aim to compare the effect on anoxic conditions on the soil capacity to adsorb P no information on the soil specific surface area (SSA) before and after the experiment is available. Moreover, I would recommend the use of SSA for isotherm to analyze the P loading on the minerals (g P/m2) while adsorption occurred.

> ***We added background soil information to Table 1, including concentrations of HCl-extractable Fe(II) and Fe(III) and AO-extractable Fe and Al. We agree with the reviewer that SSA or P loading data would be interesting, but were beyond the scope of this experiment. Given the high surface area of AO-extractable minerals, we consider their concentrations as a surrogate for SSA.***

We also question the use of just 4 points in each isotherm, without lower values or zero, the lack of fit statistics (r2, among others) and general lack of detail. Later the authors also rely in the lower values of PSI to analyze and compare their sorption curves, while they also refer to the PSI concentration as rate in page 6 line 19. The authors also mention several times the precipitiation of viviatine, while no evidence more than the mention of a MINTEQ simulation to the reader. Altough the information in in the supplementary section the reader is never refered to it. The details of this MINTEQ simulation are also omitted in the Methods section.

[Figure]

*For the justification of no zero or low values, please see the response and additions mentioned above for Reviewer 1. We also included model efficiency as a measure of model fit (Table S1, Page 7 Lines 23-24). We only used four points in the isotherms because 1) our anoxic treatment was limited by the size of the glove box, and we already have a large number of samples (n = 128); 2) including low P concentrations did not significantly affect the shape of isotherm, as our preliminary trials showed that samples receiving 10 and 50 mg P kg$^{-1}$ had extremely low P concentration in solution (< 0.1 mg P L$^{-1}$). We also justified our focus on the PSI values calculated at 1000 mg P kg$^{-1}$, as this P concentration is comparable to past studies and facilitated comparison with P sorption time curves measured at the same P addition level (Page 6 Lines 29-31).*

*We avoided the inappropriate terms including 'PSI rates' and 'P addition rates'. We justified the presence of vivianite using literature and also provided information on how MINTEQ simulation was conducted (Page 6 Line 16-22) and introduced the supplementary table in the main text.*

At the results, the authors describe the P concentrations added as rates, this is extremely confusing as rates refer to a quantity over time. Which would be the kinetic data. The authors also continue to discuss data from figure 1 (adsorption isotherms) while comparing p-values for the different concentrations of the isotherm while no table or figure is mentioned. The data that this refers is supplementary table 1, where the author refers to the rates as levels, yet in the paragraph no mention to this supplementary table is made.

*We no longer refer to P concentrations as rates and abandoned terms such as 'PSI rates' and 'P addition rates'. Rate was only used when discussing coefficient k of the P sorption curve (Page 6 Line 35). We also added a supplementary table (Table S1) to show the P concentrations from the isotherms and indicate statistical significance.*

In the discussion, the authors disregarded their kinetic and solubility data and support their discussion on the PSI and its correlation with AlOA and FeOA. The author does not seek to discuss the nuances a faster sorption rate at their simulated anoxic condition in the slope sites. On the other hand, the authors never discuss what minerals/solid phases could be the ones extracted by their HCl(FeII) and HCl(FeII) and how is this related to the higher P soption and rates in some soils. They base their conclusion on their solubility analysis yet this information is never related to the previous analysis. I would recommend the authors to discussing their results in comparison with the study "Sorption isotherms and kinetics of sediment phosphorus in a tropical reservoir" by Adhityan Appan, and Hong Wang; which is very similar to theirs.

> *We clarified our discussion of the results mentioned by the reviewer, including the kinetic and solubility data (Page 9 Line 13-30), faster sorption found under anoxic conditions (Page 9 Line 31 to Page 10 Line 12), and the phases of HCl-extractable Fe (Page 9 Line 33-35). We also discussed Adhityan and Appan 2000 in the text (Page 10 Line 24).*

Minor revisions

Page 12 line 9: change The to their

> *The text was revised accordingly.*

Page 12 line 10 change "due to the" to "do to the tropical soils'"

> *We changed the text to 'Due to the high P sorption capacity of tropical soils'.*